# Deciphering Adipose Tissue Extracellular Vesicles Protein Cargo and Its Role in Obesity

**DOI:** 10.3390/ijms21249366

**Published:** 2020-12-09

**Authors:** Tamara Camino, Nerea Lago-Baameiro, Aurelio Martis-Sueiro, Iván Couto, Francisco Santos, Javier Baltar, María Pardo

**Affiliations:** 1Grupo Obesidómica, Área de Endocrinología, Instituto de Investigación Sanitaria de Santiago de Compostela (IDIS), Xerencia de Xestión Integrada de Santiago (XXIS/SERGAS), 15706 Santiago de Compostela, Spain; tamara_cm_10294@hotmail.es (T.C.); neelago.18@gmail.com (N.L.-B.); 2Grupo Endocrinología Molecular y Celular, Instituto de Investigación Sanitaria de Santiago (IDIS), Xerencia de Xestión Integrada de Santiago (XXIS/SERGAS), 15706 Santiago de Compostela, Spain; aurelio.manuel.martis.sueiro@sergas.es; 3CIBER Fisiopatología Obesidad y Nutrición, Instituto de Salud Carlos III, 15706 Santiago de Compostela, Spain; 4Servicio de Cirugía Plástica y Reparadora, Xerencia de Xestión Integrada de Santiago (XXIS/SERGAS), 15706 Santiago de Compostela, Spain; ivan_couto@hotmail.com; 5Servicio de Cirugía General, Xerencia de Xestión Integrada de Santiago (XXIS/SERGAS), 15706 Santiago de Compostela, Spain; ffsantosbenito@gmail.com (F.S.); javier.baltar.boileve@sergas.es (J.B.)

**Keywords:** adipose tissue, extracellular vesicles, exosomes, obesity, biomarkers

## Abstract

The extracellular vesicles (EVs) have emerged as key players in metabolic disorders rising as an alternative way of paracrine/endocrine communication. In particular, in relation to adipose tissue (AT) secreted EVs, the current knowledge about its composition and function is still very limited. Nevertheless, those vesicles have been lately suggested as key players in AT communication at local level, and also with other metabolic peripheral and central organs participating in physiological homoeostasis, and also contributing to the metabolic deregulation related to obesity, diabetes, and associated comorbidities. The aim of this review is to summarize the most relevant data around the EVs secreted by adipose tissue, and especially in the context of obesity, focusing in its protein cargo. The description of the most frequent proteins identified in EVs shed by AT and its components, including their changes under pathological status, will give the reader a whole picture about the membrane/antigens, and intracellular proteins known so far, in an attempt to elucidate functional roles, and also suggesting biomarkers and new paths of therapeutic action.

## 1. Introduction

In the context of the alarming rise of obesity worldwide, there is an increasing interest to get a deeper knowledge about new signals secreted by adipose tissue at both healthy physiological and pathological status since energy imbalance causes an expansion of this tissue by hyperplasia and hypertrophy of adipocytes throughout the body. It is well known that adipose tissue is an endocrine organ able to communicate at a local or distant level with other tissues including at central level by interacting with appetite controlling areas at the hypothalamus through secreted molecules named adipokines. Since the discovery of leptin as the first adipokine back in the 1990s [1], numerous cytokines, chemokines, and hormone-like factors have been described; but more importantly they have been implicated in numerous processes such as appetite regulation, glucose and lipid metabolism, blood pressure regulation, and inflammation [2]. Thus, the alteration on the secretion of these molecules is closely related to inflammation state, impaired adipocyte metabolism, and also linked to obesity and its comorbidities. Nevertheless, the whole picture of how adipokines and cytokines participate in obesity-related diseases and how these signaling molecules vary depending on the adipose tissue anatomical accumulation is still not clearly known. Given this scenario, other alternative ways of communication have been lately revealed for adipose tissue including non-classical secretion of proteins [3], release of extracellular vesicles [4], or long non-coding [5] and microRNAs [6] shedding that may travel free or inside the mentioned extracellular vesicles.

Extracellular vesicles (EVs) are small round shaped membrane spheres of diverse size (30–1000 nm); thus, according to their size and biogenesis, EVs include: microvesicles (100 nm–1um) derived by blebbing of the plasma membrane, exosomes (30–100 nm) assembled in multivesicular endosomes (MVEs) that are secreted by exocytosis, and larger vesicles (50–5000 nm) that comprise apoptotic bodies released by cells preceding apoptosis [7,8]. Interestingly, EVs are actually shed by all cell types [9]; as a result, these vesicles can be detected in the extracellular surroundings, and virtually in every body fluid including blood, urine, cerebrospinal fluid, saliva, tears, etc. Because exosomes, microvesicles, and other EVs contain membrane and cytosolic components such as proteins, lipids, and RNAs, and this composition is conditioned by the site of biogenesis [10], they have emerged as a good source of disease biomarkers with also a potential use to monitor pathology including the effectiveness of treatments. Moreover, EVs, as essential transporters of cellular information due to their bioactive cargo (lipids, proteins, mRNAs, miRNAs), they can modify the activity or properties of specific target cells and tissues. The interaction of EVs with target cells have been proved to exert stimulatory or inhibitory functional outcomes such as cell proliferation, apoptosis, cytokine production, immune modulation, or metastasis among others by inducing cell transduction signals or by inducing genetic or epigenetic changes [11,12]. However, the great challenge surrounding EVs now, apart on deciphering the exact functional role and characterizing their dynamic content, is to find a feasible, accurate, and reliable method to analyze these vesicles for research and clinical applications [13]. There are several established procedures to isolate EVs based on differential centrifugation protocols, density gradient centrifugation, size exclusion chromatography (SEC), immunoaffinity, and polymer-based precipitation [14]; however, they all have their inconveniences, most of them being tedious or showing high variability on vesicle yield. Particularly, on studying vesicles proteome, it has to be considered that immunoaffinity and precipitants may contaminate the sample or impede protein analysis; being differential centrifugation and SEC the most useful methods for this purpose. Moreover, there is an issue with co-isolated soluble proteins/artefacts that may distort the desired analysis. Above all, it should be mention that isolation, quantification and characterization of vesicles by immunocapturing vesicle specific tretraspanins (CD9, CD63, and CD81) may give an incorrect view and distort or not represent reality. Since the secretion of vesicles is dynamic, including their content, tretraspanins may change throughout time and following physiological and pathological deregulation [15]. Unfortunately, none of the current methods is actually adapted to a routine clinical setting [16,17].

Particularly, contrary to other tissues, there is scarce knowledge about EVs shed by adipose tissue as a whole, and its individual cell components. However, the EVs have been suggested as key players in the communication between and within metabolic organs in physiological homoeostasis, and also contributing in the metabolic deregulation related to obesity, diabetes, and associated comorbidities [18,19,20]. In this regard, there are already very fascinating reports showing the participation of white AT secreted vesicles on insulin signaling in liver and muscle cells [21]. These studies show how EVs take part in a reciprocal pro-inflammatory loop between adipocytes and macrophages aggravating local and systemic insulin resistance (IR) [22,23,24,25], inducing transforming growth factor beta pathway deregulation in hepatocytes [26], attracting macrophages [27], regulating appetite and weight at central level [28], or even participating in obesity-related cancer [29,30,31]; reviewed deeply elsewhere [32]. Interestingly, the secretion of exosomes by brown adipose tissue has been also described, especially upon BAT activation [33]. Moreover, Chen and collaborators show that BAT exosomes carry the miR-92a into the circulation whose concentration is inversely correlated with human BAT activity.

It is of interest to get a better picture of specific EVs for each AT depot according to its anatomical location since there is a clear role of body fat distribution in the metabolic complications of obesity; visceral AT being the one considered most deleterious [34]. Moreover, there is a need to get a better picture of those specific vesicles secreted by the different AT cell components including immune cells that invade this tissue in the development of obesity. Additionally, there is an interest to study how these vesicles permit intercellular cross talk that includes inflammation signals and the interaction of AT cells with the extracellular matrix (ECM), which is characteristically affected in obese AT liberating ECM stiffness-promoted signals [19,35].

Therefore, the purpose of this review is gathering all the investigations performed so far around the EVs secreted by adipose tissue, and especially in the context of obesity, focusing to its protein content (Figure 1). Thus, we have performed a compilation of the most relevant reports describing EVs protein content published to date, using the following keywords: extracellular vesicles, exosomes, adipose tissue, obesity, metabolism, and obesity comorbidities.

Indeed, the better knowledge about the antigens and intracellular proteins liberated by AT-derived EVs under physiological conditions, and its alteration under pathological situations, may give a better picture about this alternative way of autocrine and paracrine communication. Interestingly, AT released EVs may provide non-invasive biomarkers, and may reveal new paths of therapeutic action.

## 2. Protein Content Characterization of EVs Shed by Adipose Tissue

EVs function is determined largely on their cargo, thus, its membrane antigens and molecular load is crucial to exert a functional effect on a target cell/tissue. Under this premise, it is vital to characterize EVs composition to elucidate changes or alterations associated to the cell of origin physiology. Contrary to other cells and tissues, the composition analysis of EVs shed by adipose tissue under normal and pathological situations is still scarce. However, there are important advances in this field, and several publications have described different adipose tissue non-coding RNAs, such as exosomal miRNAs linked to long distance metabolic regulation; many of them secreted by MSCs [36,37,38,39,40,41]. To a lesser extent, adipose tissue derived EVs proteome analysis is currently under course either in those vesicles isolated from whole adipose tissue explants, or from those secreted by its individual cell components (adipocytes, MSCs, AT macrophages, etc.) (Summarized in Table 1; Figure 2).

### 2.1. EVs Isolated from Individual Adipose Tissue Cell Components

#### 2.1.1. Cultured Adipocytes

An initial report describing human adipocytes protein content was published by Kranendonk and collaborators [22]. In their manuscript, they isolated EVs from an in vitro differentiated cell line of adipocytes of Human Simpson Golabi Behmel Syndrome (SGBS), which is a complex congenital overgrowth syndrome. Those vesicles showed to carry FABP4 and adiponectin by immunoblot; moreover, a multiplex protein array assay confirmed the presence of adiponectin and immunomodulatory proteins such as tumor necrosis factor alpha (TNF-α), macrophage-colony-stimulating factor (MCSF), and retinol binding protein 4 (RBP-4); interestingly, macrophage migration inhibitory factor (MIF) was found enriched in those adipocyte-EVs [22].

In a study focused on the link of obesity with cancer, Lazar and collaborators performed a proteome analysis of exosomes from a murine adipocyte cell line (3T3-F442A) differentiated in vitro. Functional classification of the 324 different identified proteins showed that the most represented cellular process was metabolism and transporters, being the majority implicated in lipid metabolism such as those involved in FAO: ECHA (α subunit of the trifunctional enzyme) and HCDH (hydroxyacyl-coenzyme A dehydrogenase) [31]. Later, Durcin et al., performed a detailed proteome analysis characterization of small (sEVs) and large (lEVs) vesicles secreted by a similar murine adipocyte cell line (3T3-L1) [43]. Isolated vesicles showed EV markers such as tetraspanins (CD9, CD63, CD81), Alix, lactadherin, caveolin-1 and flotillin-2, being caveolin-1 and flotillin-2 preferentially in large EVs. On the other hand, sEVs were enriched in Alix, the endosomal sorting complexes required for transport I (ESCRT-I) component, TSG101 and the tetraspanins CD9, CD63, and CD81. Interestingly, each adipocyte EV population was characterized by a specific protein profile; thus, a quantitative label-free proteomics analysis showed the presence of 533 different proteins among small and large adipocytes-EVs. A GO enrichment classification of identified proteins showed that large EVs are enriched in membrane, organelles and cell part components in comparison with sEVs. Small vesicles were instead enriched in ECM, macromolecular complex components, related to cell adhesion and also in macrophage activation. Moreover, there was an enrichment of proteins involved in metabolic pathways, most of them implicated in mitochondrial function, in both types of vesicles. Finally, and no less important, in this manuscript they select specific protein markers characteristic of adipocyte lEVs: FABP4/aP2, 14-3-3, annexin A2, endoplasmin, and actinin-4 and of sEVs: MVP, FAS, and adiponectin. Syntenin-1 was equally found in both types of vesicles. The presence of CD63, CD9, Mfg8, Flotilin-2, and Caveolin-1 was further validated in small and large vesicles secreted by primary visceral murine adipocytes by immunoblot. This information, although it has to be translated to human adipocytes, becomes very useful to characterize distinct vesicle subpopulations in the future [43].

On the pathological side, the characterization of adipocyte-derived exosomes from adipocyte primary cell cultures of obese diabetic and non-diabetic rats by Lee and collaborators, showed a different protein profile in vesicles according to the type of pathological AT. Thus, they show that EVs shed by AT from rats with insulin resistance, obesity, hypertension, hyperinsulinemia, and hyperglycemia (OLETF) have elevated presence of caveolin, lipoprotein lipase, and AQ7 compared to those vesicles from control animals characterized by AK2, catalase, and liver carboxylesterase [44]. Interestingly, the protein profile of isolated vesicles represented perfectly the expression levels on the adipocytes of origin. In this regard, our group has performed a qualitative and quantitative label-free analysis of EVs secreted by murine adipocytes in culture (differentiated C3H10T1/2 cells), that also included the analysis of those vesicles shed by insulin resistant (IR) and lipid hypertrophied (palmitate/oleic) cell models [15]. Thus, in this work, we show that the protein cargo of EVs liberated by mature adipocytes is modified and altered by metabolic insults. This analysis showed a great variation in the number of proteins identified on those vesicles secreted by the same cells but under different metabolic status. First, is worth to highlight, as previously described by others, the identification of known adipocyte associated proteins and adipokines on the EVs of mature adipocytes compared to those undifferentiated such as adiponectin, caveolae-associated protein 1, fatty acid synthase, endoplasmin-Grp94, Grp75, caveolin 1/2, adipocyte enhancer-binding protein 1, adipocyte plasma membrane-associated protein, acetoacetyl-CoA synthetase, perilipin1/4, lipoprotein lipase, major vault protein, macrophage migration inhibitory factor (MIF), fatty acid-binding protein 4 and 5 FABP-4/5, glycerol-3-phosphate dehydrogenase, hormone sensitive lipase, chemerin, and Glut1, among others. Consequently, this confirms that EVs are good representatives of the cell/tissue of origin including its metabolic/pathologic status. Accordingly, EVs from hypertrophied and IR adipocytes, showed to transport proteins previously associated to obesity and to IR such as calreticulin, S100A6, mimecan, PARK7/DJ1, PPIB, and tenascin. A protein map of each vesicle type according to the pathological situation is described, and the comparison of identified proteins with previous works on other murine adipocytes and human primary non-obese adipocytes cell cultures was performed [15]. Additionally, a quantitative label-free analysis was carried out that permitted to find protein quantity variation; by this means, a list of suggested EV biomarkers for pathological adipocytes was created (included in Table 1; Figure 2). Further validation by immunoblot confirmed that EVs shed from palmitate and oleic acid treated adipocytes were characterized by a high presence of ceruloplasmin, mimecan, and perilipin 1 adipokines, and that those vesicles from the IR cell model elicited an exclusively striking presence of the adiposity and IR related transforming growth factor-beta-induced protein ig-h3 (TFGBI) (Figure 2).

In addition to those studies in murine adipocyte cell models, other reports have shown the protein content of EVs liberated by human cultured adipocytes. A proteomics analysis of EVs secreted form primary human subcutaneous preadipocytes differentiated to adipocytes, described 884 proteins named exo-adipokines that contribute to the AT secretome; many of them were related to signaling pathways of membrane-mediated processes that suggest their participation in inter-organ crosstalk [42].

#### 2.1.2. Cultured Adipose-Derived Mesenchymal Stem Cells (ADSCs)

A comparative proteomic analysis of extracellular vesicles isolated from porcine adipose tissue-derived mesenchymal stem/stromal cells was published by Eirin et al., 2016 [45]. In this work, the authors describe that proteins enriched in EVs are linked to a large variety of biological functions such as angiogenesis, blood coagulation, apoptosis, extracellular matrix remodeling, and regulation of inflammation. Thus, they suggest that EVs have a selectively-enriched protein cargo with a specific biological signature that ADSCs may utilize for intercellular communication to ease tissue repair. Additionally, a proteomics analysis of the exosomes secreted by ADSCs to elucidate their potential role as therapeutic strategy for tissue injury has been published [46]. In their work, they mention 1185 protein groups, many of them participating in metabolic pathways, focal adhesion, regulation of actin cytoskeleton, microbial metabolism, and more interesting, some belonged to tissue repair-related signaling pathways.

### 2.2. EVs Secreted by Whole Adipose Tissue Explants

The analysis of EVs secreted by whole AT explants has disadvantages since it may be technically complicated to handle, and may involve a higher risk of artefacts or contaminants. However, the isolation of EVs from whole fresh AT in culture is of interest because it comprises not only those vesicles secreted by mature adipocytes, but also those liberated by the stromal vascular fraction (SVF) including pre-adipocytes, endothelial cells, and also innate and adaptive immune cells such as macrophages and lymphocytes. This approach, may indeed give a more physiological and closer view to AT reality as the 3D structure of the tissue is maintained, and also the intercellular cross talk is retained. The former includes inflammation signals and the interaction of cells with the extracellular matrix (ECM) which is characteristically affected in obesity-deregulated AT including ECM stiffness-promoted signals [35].

This approach was selected by Kranendonk and collaborators to characterize the adipokine profile of EVs liberated by human explants of subcutaneous and omental AT collected from lean subjects undergoing surgery for aneurysmatic aortic disease [21]. By multiplex immunosassay, they described not only the presence of six known adipokines (IL-6, MIF, MCP-1, adiponectin, resistin, and RBP-4) carried by AT secreted vesicles, but their variation according to the tissue of origin. Thus, vesicles from omental AT showed higher concentration than subcutaneous of IL-6, MIF, and MCP-1 [21].

An interesting study analyzed the AT exosomal proteomic profile under the context of the maternal and fetal communication [47]. Thus, they show a quantitative proteome analysis of exosomes isolated from cultured omental AT secretomes of normal glucose tolerant (NGT) pregnant women compared to those with gestational diabetes mellitus (GDM). Besides they observe that the number of vesicles was significantly greater in GDM than in NGT, the proteome study shows a differential expression of the proteins targeting the sirtuin signaling pathway, oxidative phosphorylation, and mechanistic target of rapamycin signaling pathway in GDM in relation to NGT [47]. More recently, others have identified novel adipokines through proteomic profiling of small EVs isolated from rat inguinal adipose tissue. In this work, they describe three novel vesicle-adipokines, NPM3, DAD1, and STEAP3, whose expression is altered in obese animals compared to lean [49].

To the best of our knowledge, our group was pioneer on analyzing EVs protein cargo liberated by whole human obese visceral (VAT) and subcutaneous (SAT) adipose tissue explants [48]. This study is informative because it describes striking protein composition differences between vesicles secreted by VAT and SAT, paralleling the biological and endocrine behavior of both tissues. Overall, 574 different proteins were described for VAT vesicles, and 401 for SAT, and the most relevant constituents classified according to their function were represented in obese VAT and SAT EV reference maps. Apart from a greater number of different proteins, obese VAT vesicles contained more adipose tissue and obesity-related proteins than SAT vesicles. Therefore, it is of interest the identification of adipokines such as leptin, GRP78, hormone sensitive lipase, ceruloplasmin, DDP-4, and septin 11, among others, in vesicles isolated from obese VAT. According to functional classification, EVs from both depots carried signaling proteins from TRAIL pathway, VEGF and VEGFR, and syndecan-1 mediated; also, cell surface interaction integrins, and proteins related to proteoglycan syndecan-mediated signaling events were found, especially in those vesicles from subcutaneous obese adipose tissue. SAT EVs showed a higher percentage than those in VAT in terms of extracellular matrix (ECM) constituents, signal transduction, and cell growth, maintenance and communication; and on the contrary, VAT vesicles evidenced more proteins related to energy pathway and metabolism and immune system process. A MS-label free quantitative analysis (SWATH) permitted to discern that those vesicles from visceral AT exhibit an enrichment of proteins implicated in AT inflammation and insulin resistance such as TGFBI, CAVN1, CD14, mimecan, thrombospondin-1, FABP-4, or AHNAK. On the other hand, syntenin-1, fibrilin-1, CD136, ALIX, CD98, and others were elevated in vesicles from SAT. Amusingly, in this analysis, we found many proteins common to those described by us previously in EVs from murine insulin resistant and lipid hypertrophied (palmitate and oleic acid) C3H10T1/2 adipocytes [15]; 38%, 32.5%, and 20% of human VAT and SAT EVs proteins were common to those identified in EVs from insulin resistant, palmitate, and oleic acid hypertrophied adipocytes respectively [48]. Further immuno-validation assays that included also vesicles from lean control adipose tissues, confirmed that obese VAT vesicles are characterized by a diminution of syntenin 1 and the elevation of TFGBI and mimecan.

### 2.3. Circulating EVs in Obesity

There is now strong evidence about the existence of adipose tissue secreted vesicles at circulating level [55,56]; therefore, these vesicles are now linked to metabolic signaling as a very sophisticated and accurate alternative system of cell and tissue/hormonal communication. Importantly, EVs released by AT, have been shown to participate on inter-organ communication permitting AT to send messages to other periphery tissues (liver, muscle, pancreas, etc.), and also at central level through the food intake regulating areas in the brain since it has been described their ability to cross the blood-brain barrier [57]. Moreover, it is now clear that EVs shed by adipose tissue may exert a crucial role in obesity and its associated comorbidities including type 2 diabetes, vascular disease, liver steatosis, inflammation, and cancer, among others, becoming an attractive source of disease biomarkers at circulating level [19,55,58,59,60]. Of interest, is the recent and exponential description of different EVs derived miRNAs that exert post-transcriptional regulation of mRNAs in target cells/tissues at both physiological and pathological level revised elsewhere [40,61].

Several data demonstrated the presence of AT shed EVs at circulating level; thus, it was described the presence of adiponectin and trace amounts of resistin in exosomal vesicles isolated in the serum of mice after ultracentrifugation [54]; moreover, adiponectin was described in EVs from human plasma [21].

In relation to cardiovascular disease (CVD), an interesting work assesses the relationship between AT quantity, AT distribution, and metabolic parameters of AT (dys)function towards plasma levels of four CVD-associated EV-markers. Also, they investigated the relation between those EVs-markers and metabolic syndrome, or incident type 2 diabetes in patients with vascular disease. They conclude that EV-cystatin C was positively related to metabolic complications of obesity, including low-grade systemic inflammation, low HDL-cholesterol levels, and metabolic syndrome. On the contrary, EV-CD14 was inversely related to AT abundance and dyslipidaemia, and was in addition related to a relative risk reduction for the development of type 2 diabetes [50].

In a model of murine obesity, it was described that the production of circulating EVs is increased during obesity and correlated to glucose intolerance and macrophage infiltration in AT; and more interestingly, the authors describe perilipin A (perilipin-1) as an adipose tissue EV biomarker that can be detected by immunoblot at circulating level. In this manuscript, they prove that circulating EVs-perilipin increases in murine and human obesity, being also correlated with insulin resistance. Moreover, perilipin A quantity in circulating EVs was significantly diminish after calorie restriction [56]. Furthermore, these same authors extend those findings by analyzing EVs in a cohort of 203 subjects with or without risk factor for metabolic diseases. Their results show various interesting facts: that the circulating EV number is significantly higher in men than in women, and higher in those with impaired oral glucose tolerance test. Besides, they find a correlation between the number of circulating EVs, the elevated TG content, and the homeostasis model assessment-β-cell function (HOMA-β) value [51].

Precisely, in relation to obesity, Amosse and co-authors have shown the phenotyping of general circulating EVs in plasma collected from metabolic syndrome patients [52]. They describe that microvesicles (MVs) and exosomes increase significantly with body mass index (BMI). Moreover, by commercial protein arrays, they reveal that those vesicles contain AT-derived adipokines. Precisely, they discovered that half of the plasma macrophage migration inhibitory factor (MIF) circulates associated to MVs conserving its functional activity on macrophages. Although it cannot be discerned if those analyzed vesicles had AT origin (MVs relies on a specific increase in platelet and endothelium-derived MVs), they may synergy with AT vesicles on the development of pathology. Iwitczak and collaborators analyzed circulating general EVs paying special attention to adipocyte markers (adiponectin, FABP4, and PPARγ) in obese patients before and after bariatric surgery [53]. In this study, they find that EV-FABP4 increases at 1 month after surgery, returning to baseline by 6 months paralleling those levels of the soluble circulating protein level. Moreover, they describe that those patients who underwent biliopancreatic diversion showed less FABP4-EVs at 6 months compared to those that underwent sleeve gastrectomy/gastric banding even though both groups of patients lost a similar amount of weight. They conclude that those changes in circulating EV- and plasma-derived FABP4 after bariatric surgery may represent alterations in adipose tissue homeostasis.

Finally, is worth highlighting our recent investigations on describing the proteome of human obese AT secreted EVs mentioned above [48]. In this work, we were able to detect candidate obesity and insulin resistance (IR) biomarker TFGBI by ELISA on EVs isolated at circulating level in plasma of obese and lean individuals. Importantly, the amount of EV-TFGBI corrected towards EV-caveolin, as an AT-specific protein, has revealed that those obese patients with a history of insulin resistance showed a statistically significant elevation of this protein on their circulating vesicles compared to the non-IR obese patients, and to lean individuals.

## 3. Conclusions

At this moment, the extracellular vesicles are acquiring enormous relevance due to their signaling role, as a vector carrying biomarkers of the cell of origin including its pathological deregulation, and also by providing new therapeutic options. Precisely, the role of EVs released by endocrine tissues, took certain time to emerge compared to other cellular systems such as in the immune system or in pathologies such as cancer; however, the development of investigations regarding EVs in the context of metabolic regulation has increased significantly in the last years. Consequently, there is currently no doubt about the potential applications of EVs to get a deeper insight about adipose tissue communication at local and peripheral/central level.

In this review, we tried to compile all the information up to date in relation to adipose tissue EVs specific protein content, and its variation under metabolic stress and/or in the context of obesity and its comorbidities. There are now several reports describing AT EVs-ARNs [62,63]; conversely, the description of EVs protein cargo, has not paralleled those studies. In this review we describe the protein content of EVs isolated from human and animal adipocytes in culture, from the stromal vascular fraction, and also from whole adipose tissue paying special attention to those proteins elevated during the development of obesity. We can conclude that adipose tissue reflects its metabolic status and deregulation through its secreted vesicles, which carry not only adipocyte cell makers including well known adipokines such as leptin or adiponectin, but also markers of malignancy. The investigations done so far have shown that obesity is characterized by a greater number of EVs secretion that it is paralleled at circulating level. Moreover, it can be affirmed that those adipocytes under metabolic insults (insulin resistant or lipid hypertrophied) or obese AT shed a greater number of EVs whose type and cargo depends on its anatomical origin. Thus, in relation to protein content, those vesicles secreted by visceral AT have shown a greater number of different and adipose-related proteins than subcutaneous; accordingly, those visceral vesicles carry a greater number of obese and comorbidities associated proteins. Importantly, in this review we draw attention to protein perilipin 1 as an AT derived-EVs specific protein that may allow to discern adipocyte specific vesicles among others at circulating level [15,48,56]. More interestingly, we compiled the most representative EVs proteins of pathological AT tissue that may represent good circulating biomarkers such as the perilipin itself, cystatin C, FABP-4, mimecan, and TFBI.

## 4. Future Perspectives

There is now a need to get a broader picture on the functional role of AT-derived EVs including its target cells/tissues, and to validate the presence of those vesicles in a reasonable cohort of patients to confirm their value as metabolic biomarkers. A better knowledge of the functional role of EVs released by tissues altered in obesity (white and brown adipose tissue, inflamed macrophages, muscle, liver, etc.), will probably allow identifying possible therapeutic targets for the treatment of obesity and associated diseases. Moreover, the selection of a panel of EV biomarkers could have a great application to the clinic, being a very useful tool in the obesity consultation. Those vesicles, easily analyzed at a circulating level in a non-invasive way, would be indicative of type/amount of accumulated fat (visceral vs. subcutaneous), impaired adipocyte metabolism, obesity treatment efficiency (diet/drugs/surgery), inflammation level, or appearance and monitoring of comorbidities (T2D, fatty liver, etc.). Hence, EVs provide a wide range of new possibilities both for better understanding the deregulation of signals in the development of obesity, and probably providing new therapeutic targets, and for monitoring the disease.

## Figures and Tables

**Figure 1 ijms-21-09366-f001:**
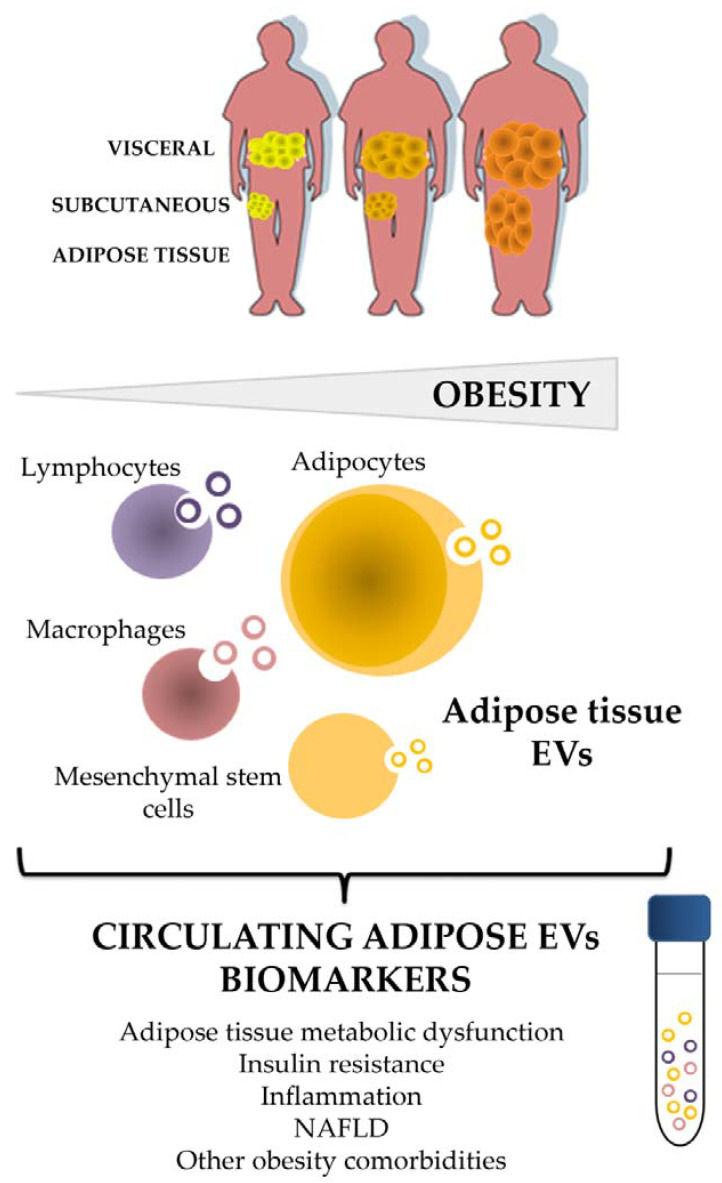
Adipose tissue and its cellular components shed EVs as an alternative way of communication at paracrine and endocrine level that reflects characteristics of the metabolic status, and emerge as a biomarker source of disease. EVs: extracellular vesicles; NAFLD: non-alcoholic fatty liver disease.

**Figure 2 ijms-21-09366-f002:**
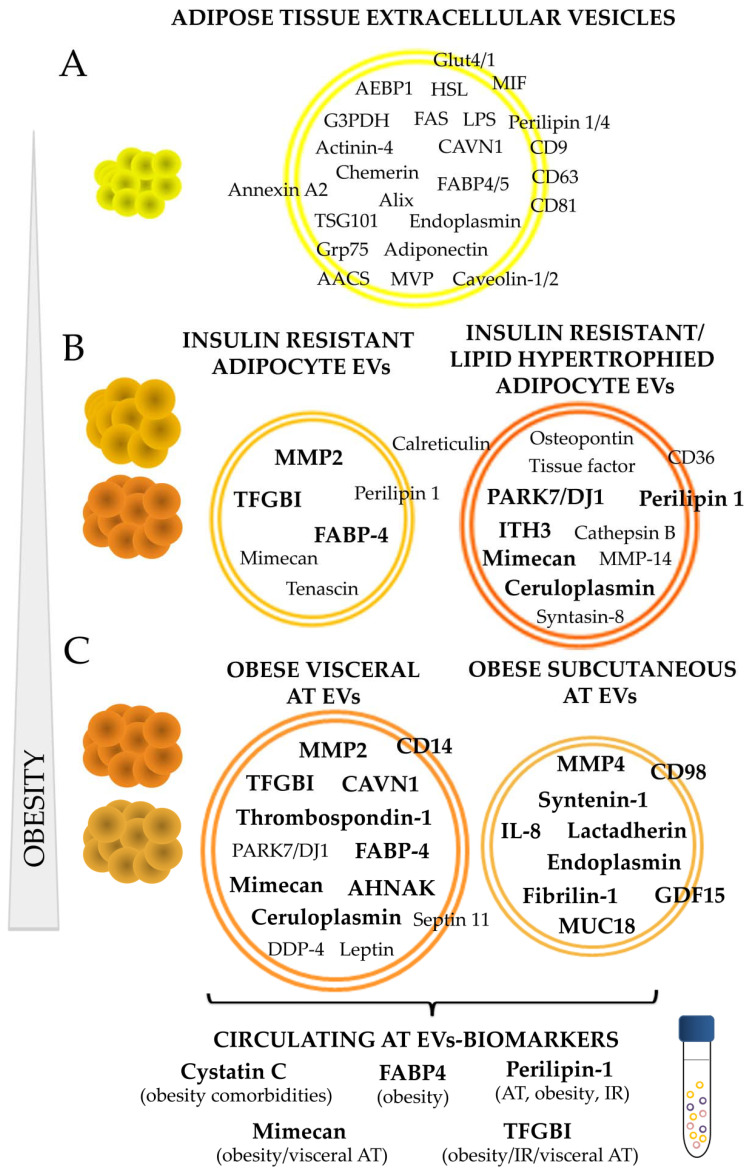
Adipose tissue EVs protein cargo and its alteration on the course of obesity. A summary of AT specific proteins described so far in the literature including those proteins present in EVs shed by normal/healthy adipocytes (**A**); those present or elevated with insulin resistance, lipid hypertrophied adipocytes (**B**); and those characteristic of subcutaneous and visceral human obese AT (**C**). Proteins in bold were found elevated compared to control healthy adipocytes, or compared to those released by adipose tissue of lean individuals. AT: adipose tissue; IR: insulin resistance; AEBP1: adipocyte enhancer-binding protein 1; HSL: hormone sensitive-lipase; FAS: fatty acid synthase; MVP: major vault protein; MIF: macrophage migration inhibitory factor; AACS: acetoacetyl-CoA synthetase; LPS: lipoprotein lipase; FABP: fatty acid binding protein; G3PDH: glyceraldehyde 3-phosphate dehydrogenase; CAVN1: caveolae associated protein 1; MMP: matrix metalloproteinase; ITH3: inter-α-trypsin inhibitor heavy chain H3; GDF15: growth differentiation factor 15; IL: interleukin; MUC18: cell surface glycoprotein; AHNAK: neuroblast differentiation-associated protein; TGFBI: transforming growth factor-beta-induced protein ig-h3.

**Table 1 ijms-21-09366-t001:** Overview of revised publications related to adipose tissue EVs protein cargo characterization. Reference, method of EV isolation, type of protein analysis, type of cell/tissue or circulating analysis, species, and propose EVs markers are shown. dUC: differential ultracentrifugation; SDG: Sucrose density grandient; UF: ultrafiltration; TEIR: total exosome isolation reagent; LC-MS/MS: liquid chromatography coupled to mass spectrometry; SWATH: sequential window acquisition of all theoretical mass spectra; ELISA: enzyme-linked immunosorbent assay; MV: microvesicles; EXO: exosomes; CRP: C-reative protein; ICAM-1: intercellular adhesion molecule 1; DDP-4: dipeptidyl peptidase-4; GDM: gestational diabetes mellitus; MSCs: mesenchymal stem cells; VAT: visceral adipose tissue; SAT: subcutaneous adipose tissue.

Ref	Isolation Method	Type of Protein Analysis	Cell/Tissue Type/Circulating	Species	EVs Markers
**Cultured adipocytes**					
Kranendonk et al., 2014 [22]	dUC	Immunoblot + multiplex immunoassay	Simpson Golabi Behmel Syndrome (SGBS) adipocytes cell line	Human	FABP4, adiponectin, TNF-α, MCSF, RBP4, MIF
Hartwing et al., 2019 [42]	UF + dUC	NanoLC-MS/MS	Human primary adipocytes from lean to moderate overweight woman	Human	Adiponectin, FABP4, metalloproteases
Lazar et al., 2016 [31]	SDG	NanoLC-MS/MS	3T3-F442A adipocyte cell line	Mouse	ECHA, HCDH (FAO INVOLVED)
Durcin et al., 2017 [43]	dUC	NanoLC-MS/MS	3T3-L1 adipocyte cell line	Mouse	L-Evs: FABP4, 14-3-3, Annexin A2, endoplasmin, actinin-4 sEVs: MVP, FAS, adiponectin
Camino et al., 2020 [15]	dUC	Nano-LC/MS-MS + SWATH	C3H10T1/2 adipocyte cell line: models of IR and lipid hypertrophy	Mouse	Ceruloplasmin, mimecan, perilipin 1, TFGBI
Lee et al., 2015 [44]	dUC	NanoLC-MS/MS+ label-free	Obese diabetic and obese nondiabetic adipocytes of Otsuka Long-Evans Tokusima Fatty (OLETF) primary cell culture	Rat	Caveolin, LPL, AQ7
**Cultured adipose-derived mesenchymal stem cells**					
Eirin et al., 2016 [45]	dUC	LC-MS/MS	MSCs from abdominal fat	Porcine	C2, VEGF, vWF (vonWillebrand factor)
Xing et al., 2020 [46]	dUC	LC-MS/MS	Adipose-derived mesenchymal stem cells (ADSCs)	Mouse	Tissue repair
**Adipose tissue explants**					
Kranendonk et al., 2014 [22]	dUC	Adipokine profile array	Subjects undergoing surgery for aneurysmatic aortic disease (obese/overweight)	Human	MCP-1, IL-6, and MIF in omental AT vs. subc
Jayabalan et al., 2019 [47]	dUC	LC-MS/MS(SWATH)	Omental adipose tissue from woman with GDM after baby delivery	Human	Proteins related to glucose metabolism
Camino et al., 2020 [48]	dUC	NanoLC-MS/MS (SWATH)	Visceral and Subcutaneous whole AT explants from obese	Human	VAT: TGFBI, CAVN1, CD14, mimecan, thrombospondin-1, FABP-4,AHNAK/↓Syntenin 1
Zhang Y et al., 2020 [49]	TEIR	LC-MS/MS	Inguinal fat pads obese Sprague-Dawley rats	Rat	NPM3, DAD1and STEAP3
**Circulating EVs**					
Kranendonk et al., 2014 [50]	ExoQuick	Multiplex immunoassay	Circulating:plasma patients with vascular disease	Human	Cystatin C positive related to obesity metabolic complications/CD14 negative related
Kobayashi et al., 2018 [51]	qEV column	Immunoblot	Circulating: plasma patients with metabolic disease	Human	Perilipin 1 elevated with metab risk factors
Amosse et al., 2018 [52]	dUC	Protein arrays/multiplex	Circulating MVs and EXO plasma from metabolic syndrome patients	Human	Adiponectin, adipsin, cathepsin D, CRP, ICAM-1 chemerin, DDP-4 among others.
Witczak et al., 2018 [53]	dUC	Immunoassay	Circulating plasma obese before and after bariatric surgery	Human	FABP4
Phoonsawat et al., 2014 [54]	dUC	Immunoblot/ELISA	Circulating: serum obese mice	Mouse	Adiponectin, resistin

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
