# Peer review of "Deciphering Adipose Tissue Extracellular Vesicles Protein Cargo and Its Role in Obesity"

_ijms, 2020, doi:10.3390/ijms21249366_

Round 1

Reviewer 1 Report

The review article by Camino et al entitled “Deciphering adipose tissue extracellular vesicles protein cargo and its role as biomarkers for monitoring obesity” is overall well written and comprehensive from leaders in the field.

A few points to further achieve excellence:

Is anything known about the role of EVs from brown adipose tissue?

Can the authors further elaborate on what “obesity-related alterations are”on Line 51?

Can the authours clarify on Line 53-55 “Given this scenario, other alternative ways of communication have been lately revealed for adipose tissue including non-classical secretion[3], release of extracellular vesicles[4], or long non-coding[5] or 55 microRNAs[6] shedding.” Is this within EVs?

Can the authours elaborate in lines Line 77-79 on what the impact the protocols have on protein cargo and what is considered the gold standard in the field?

In lines 172-174 Does the differences between large and small EVs relate to the smaller surface area affecting the enrichment of membrane proteins vs. intracellular proteins?

In lines 228-230 Is there evidence that metabolic signals control the packaging and release of EVs?

Small issues

  • The spacing between the reference number and the last word and period is not consistent.
  • Can the title be improved/simplified for clarity?
  • In figure 1, lymphocytes and macrophages are not standard English spelling.
  • Can the table be organized into studies by mouse, rat, human cells, human tissues under appropriate subheadings?
  • Figure 2 caption Proteins in bold were found elevated – Was this compared to control?
  • Section 2.2.2 mesenchymal is not the standard English spelling

Author Response

The review article by Camino et al entitled “Deciphering adipose tissue extracellular vesicles protein cargo and its role as biomarkers for monitoring obesity” is overall well written and comprehensive from leaders in the field.

 A few points to further achieve excellence:

Is anything known about the role of EVs from brown adipose tissue?

Thank you so much for bringing this issue up since it is true that it is an interesting point to be added to this review that we have missed. There is an interesting work by Chen and collaborators published in Nature communications showing that brown adipocytes release exosomes, and that BAT activation increases exosome release. Moreover, they performed miRNAs profiling in exosomes released from brown adipocytes, and in exosomes isolated from mouse serum showing that levels of miRNAs change after BAT activation in vitro and in vivo (Chen et al., Nat Communications 2016). Please note that we have added this in the new version of the review. 

Can the authors further elaborate on what “obesity-related alterations are”on Line 51?

We have corrected this expression; we meant obesity-related diseases.

Can the authours clarify on Line 53-55 “Given this scenario, other alternative ways of communication have been lately revealed for adipose tissue including non-classical secretion[3], release of extracellular vesicles[4], or long non-coding[5] or 55 microRNAs[6] shedding.” Is this within EVs?

We meant both, free and within EVs; please see that we have clarified this as requested.

Can the authours elaborate in lines Line 77-79 on what the impact the protocols have on protein cargo and what is considered the gold standard in the field?

This is a good point and useful for the readers; please see that we have added this data in the introduction section.

In lines 172-174 Does the differences between large and small EVs relate to the smaller surface area affecting the enrichment of membrane proteins vs. intracellular proteins?

This is probably true; to the best of our knowledge, there is no such analysis. However, our experience is that isolation of the same amount of vesicles, with same size, renders different amount/type of intra-vesicle and vesicle membrane proteins. We believe that the number and type of proteins inside and in the surface of vesicles are more related to the physiological/pathological status or even to the stimulus (stress) of the progenitor cell, than to other factors.

In lines 228-230 Is there evidence that metabolic signals control the packaging and release of EVs?

There is no such evidence, but there are plenty of indirect data that suggest this. Indeed our experience have shown us that the amount and protein content of extracellular vesicles change with metabolic alterations as shown in our studies culturing adipocytes in the presence of palmitate or oleic acid, or by treating those cells with high glucose and high insulin.

Small issues

  • The spacing between the reference number and the last word and period is not consistent.

This was mended.

  • Can the title be improved/simplified for clarity?

Please see that we have shortened the title as recommended.

  • In figure 1, lymphocytes and macrophages are not standard English spelling.

This was corrected

  • Can the table be organized into studies by mouse, rat, human cells, human tissues under appropriate subheadings?

We tried to organize this table by citing the studies in the same order as in the text, to be coherent and to facilitate the reading. Thus, we started with those works performed in cell lines, followed by those done in whole adipose tissue and finally, those at circulating level. However, we agree with the reviewer that subheadings will make the table more organized and easy to follow by the reader. Taking into account this suggestion, we have also ordered the data according to the species within each subheading.

  • Figure 2 caption Proteins in bold were found elevated – Was this compared to control?

Yes, we meant that they are elevated compared to lean individuals in the case of human  obese, and compared to control adipocytes (“healthy”). We have indicated this in the figure legend.

  • Section 2.2.2 mesenchymal is not the standard English spelling

This was corrected

Reviewer 2 Report

Whit great interest I read the review on a current topic that will soon have considerable feedback. I think it is useful for the authors to insert the methods of choice of the selected papers. I understand that it is not a systematic review. But sharing keywords, filters and the time interval could be useful. Figure number two is very important, but it could be made less confusing by better ordering the molecules. I would divide future perspectives from conclusions. Clear examples of treatment hypotheses or diagnostic systems could be added in future perspectives. The conclusions could contain the concepts that are explained in Figure 2, which in my opinion are very useful. Some sentences are long and should be shortened. After this the papero could be accepted.

Author Response

Whit great interest I read the review on a current topic that will soon have considerable feedback. I think it is useful for the authors to insert the methods of choice of the selected papers. I understand that it is not a systematic review. But sharing keywords, filters and the time interval could be useful.

Thank you so much for the carefully reading and the feedback. We agree with the reviewer in this comment; please see that we have added this information at the end of the introduction section in the new version of the manuscript.

Figure number two is very important, but it could be made less confusing by better ordering the molecules.

Please see that we have ordered Figure 2 as requested.

I would divide future perspectives from conclusions. Clear examples of treatment hypotheses or diagnostic systems could be added in future perspectives. The conclusions could contain the concepts that are explained in Figure 2, which in my opinion are very useful. Some sentences are long and should be shortened. After this the paper could be accepted.

Please see that we have divided conclusions and future perspectives as requested, highlighting the traslacional potential of the described findings to the clinic.